# Consistent View Alignment Improves Foundation Models for 3D Medical Image Segmentation

**Puru Vaish[1]***    **Felix Meister[2]**    **Tobias Heimann[2]**    **Christoph Brune [1]**    **Jelmer M. Wolterink [1]**

[1] Department of Applied Mathematics, Technical Medical Centre, University of Twente
[2] Digital Technology and Innovation, Siemens Healthineers, Erlangen, Germany
`{p.vaish, c.brune, j.m.wolterink}@utwente.nl`
`{felix.meister, tobias.heimann}@siemens-healthineers.com`

## Abstract

Many recent approaches in representation learning implicitly assume that uncorrelated views of a data point are sufficient to learn meaningful representations for various downstream tasks. In this work, we challenge this assumption and demonstrate that meaningful structure in the latent space does not emerge naturally. Instead, it must be explicitly induced. We propose a method that aligns representations from different views of the data to align complementary information without inducing false positives. Our experiments show that our proposed self-supervised learning method, *Consistent View Alignment*, improves performance for downstream tasks, highlighting the critical role of structured view alignment in learning effective representations. The code and pretrained model weights are released at [github.com/Tenbatsu24/LatentCampus](github.com/Tenbatsu24/LatentCampus).

## 1 Introduction

Learning robust and transferable representations is a core challenge in modern machine learning. Contrastive frameworks have shown strong success across modalities by distinguishing positive and negative pairs, enabling semantically rich embeddings from both labelled and unlabelled data Oord et al. (2018); Chen et al. (2020); Caron et al. (2021); Maxime Oquab et al. (2023).

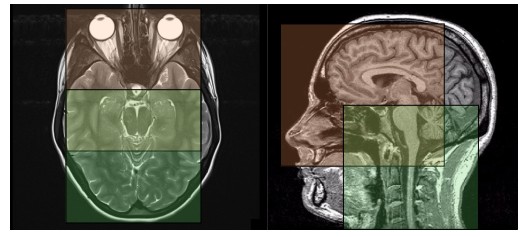

Figure 1: Two examples illustrating the issue of loosely correlated views. Although crops overlap spatially, they may represent different semantics. Existing self-supervised methods still treat them as positives, forcing misaligned features and degrading representation quality.

However, these methods rely on the assumption that positive pairs share meaningful semantic content. When this assumption breaks, such as when two augmented views are only loosely correlated, models are forced to align unrelated features, introducing spurious associations and degrading representation quality Chuang et al. (2022); Jing et al. (2022). Prior approaches mitigate this via robust losses or improved pair sampling Ghosh et al. (2015); Wang et al. (2019); Ozair et al. (2019), but they seldom control *where* in the feature space alignment occurs.

---

*Corresponding author

39th Conference on Neural Information Processing Systems (NeurIPS 2025) Workshop: MedEurIPS.

As illustrated in Fig. 1, current methods rarely enforce local, semantically consistent correspondences, leaving representations vulnerable to false-positive alignments. This motivates our central question: can alignment be explicitly regularised to occur only between truly corresponding regions?

To this end, we introduce Consistent View Alignment (CVA), a self-supervised framework that enforces spatially grounded consistency between overlapping regions of augmented views. By constraining alignment to semantically matched areas, CVA mitigates false positives and preserves meaningful latent structure, yielding more stable and generalisable representations.

## 2 Methodology

We propose Consistent View Alignment (CVA), a self-supervised framework that learns spatially consistent and transferable visual representations by enforcing feature agreement only between semantically corresponding regions across augmented views.

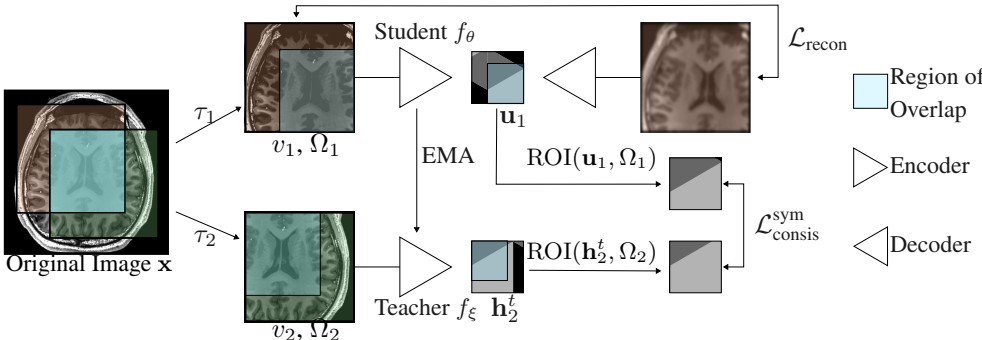

Figure 2: Overview of the Consistent View Alignment (CVA) framework. Two overlapping crops (40-80%) from the same image are encoded by student and teacher networks. The reconstruction branch uses a masked autoencoding objective, while the alignment branch matches overlapping regions via ROIAlign. A consistency loss enforces agreement only on aligned regions, reducing spurious matches and promoting spatially coherent representations.

### 2.1 Consistent View Alignment

CVA comprises two components: (1) *consistent view generation* and (2) *feature alignment with consistency loss*, as illustrated in Figure 2. **(1)** From each input image (spatially augmented), two random crops are sampled with an overlap ratio constrained between 40% and 80%, ensuring a shared region of semantic consistency. Intensity and noise augmentations are applied on the two views. The bounding boxes of these overlapping areas are recorded and later used to align features extracted. **(2)** Each view is encoded by a student-teacher pair of networks, with the teacher updated via exponential moving average (EMA). Using the stored overlap coordinates, ROIAlign extracts aligned feature patches from both views, focusing the consistency objective on semantically corresponding regions.

Let $\mathbf{u}_1^{\Omega_1}$ and $\mathbf{h}_2^{\Omega_2,t}$ denote the aligned feature maps from the student and teacher branches, respectively. Local feature consistency is enforced via a cosine regression loss (referred to as CVA) adapted from SimSiam Chen and He (2021) or a NT-Xent Chen et al. (2020) loss (referred to as C-CVA), with a symmetric formulation Caron et al. (2020) to stabilise training and remove directional bias:

$$\mathcal{L}_{\text{cos}} = 2 - 2\frac{\mathbf{u}_1^{\Omega_1} \cdot \mathbf{h}_2^{\Omega_2,t}}{\|\mathbf{u}_1^{\Omega_1}\|_2 \|\mathbf{h}_2^{\Omega_2,t}\|_2}, \quad \mathcal{L}_{\text{NT-Xent}} = -\log\frac{\exp(\text{sim}(\mathbf{u}_1^{\Omega_1}, \mathbf{h}_2^{\Omega_2,t})/\tau)}{\sum_j \exp(\text{sim}(\mathbf{u}_1^{\Omega_1}, \mathbf{h}_j)/\tau)},$$

$$\mathcal{L}_{\text{consis}}^{\text{sym}} = \tfrac{1}{2}\mathcal{L}_{\text{consis}}(\mathbf{u}_1^{\Omega_1}, \mathbf{h}_2^{\Omega_2,t}) + \tfrac{1}{2}\mathcal{L}_{\text{consis}}(\mathbf{u}_2^{\Omega_2}, \mathbf{h}_1^{\Omega_1,t}). \tag{1}$$

### 2.2 Overall Objective

The complete training objective combines reconstruction, consistency, and optionally, contrastive components:

$$\mathcal{L} = \lambda_{\text{recon}} \mathcal{L}_{\text{recon}} + \lambda_{\text{consis}} \mathcal{L}_{\text{consis}}^{\text{sym}} + \lambda_{\text{con}} \mathcal{L}_{\text{con}}^{\text{sym}}, \tag{2}$$

where $\lambda_{\text{recon}}$, $\lambda_{\text{consis}}$, and $\lambda_{\text{con}}$ are balancing weights. The reconstruction term $\mathcal{L}_{\text{recon}}$ preserves low-level image fidelity, the consistency term $\mathcal{L}_{\text{consis}}^{\text{sym}}$ enforces alignment between semantically corresponding features across views, and the optional contrastive term $\mathcal{L}_{\text{con}}^{\text{sym}}$ between pooled feature maps of student and teacher which promotes global discriminative structure in the latent space using a symmetrised NT-Xent loss. We ablate between two consistency formulations: the symmetrised cosine regression loss ($\mathcal{L}_{\text{cos}}^{\text{sym}}$, CVA) and its NT-Xent contrastive variant ($\mathcal{L}_{\text{NT-Xent}}^{\text{sym}}$, C-CVA).

## 3 Experiments and Results

**Experimental Setup.** We evaluate Consistent View Alignment on large-scale 3D MRI pretraining and multiple downstream medical imaging benchmarks. All models are pretrained on the *OpenMind* dataset Wald et al. (2025b), which contains over 110,000 head & neck MRI volumes from 34,000 patients across multiple modalities (T1w, T2w, FLAIR, FA, MD, etc.). All models were trained on one A40/L40 GPU (48GB memory). We test two representative backbones: the convolutional *ResEnc-L* Isensee et al. (2024) and the transformer-based *Primus-M* Wald et al. (2025a), covering distinct inductive biases. The teacher network is updated via EMA during pretraining with a momentum of 0.995. Pretraining follows a two-stage protocol: (1) MAE-based initialization for 1000 epochs and (2) post-pretraining using CVA or its variants (150 epochs for Primus-M and 250 for ResEnc-L). Augmentation follow standard nnUNet defined transformation with controlled overlap (40, 80%). The two stage training reduces computer burden for our ablations from 112 to 31 in GPU days.

**Downstream Evaluation.** We fine-tune models on four segmentation datasets: Yale Brain Metastisis (YBM), BraTs Post-Glioblastoma (GLI), Ischemic Stroke Lesions (ISL), Brain Tumour Segmentation from Medical Segmentation Decathlon (MSD) and one classification task (ABIDE II, ASD vs. control). Segmentation performance is reported as the mean of Dice (DSC) and Normalised Surface Dice (NSD) (1 mm), while classification uses balanced accuracy, AUROC, and average precision, averaged across folds. All fine-tuning takes place for 150 epochs and 250 iterations per epoch.

Table 1: Comparison of segmentation and classification performance across reconstruction and consistency variants. Lower ranks indicate better performance.

| Track | Recon. | Consis. | Cont. | Avg Rank | Seg Rank | Cls Rank | ISL DSC | ISL NSD | YBM DSC | YBM NSD | GLI DSC | GLI NSD | MSD DSC | MSD NSD | ABD II Bal Acc. | ABD II AUROC | ABD II AP |
|---|---|---|---|---|---|---|---|---|---|---|---|---|---|---|---|---|---|
| | AE | ✗ | ✗ | 6.08 | 6.63 | 5.00 | 77.34 | 75.57 | 60.92 | 69.44 | 68.38 | 73.41 | 72.66 | 76.64 | 57.30 | 60.61 | 60.03 |
| | MAE | ✗ | ✗ | 5.22 | 5.00 | 5.67 | 78.87 | 76.66 | 61.21 | 68.68 | 69.83 | 75.02 | 72.22 | 76.49 | 57.33 | 60.18 | 58.89 |
| | MAE | CVA | ✗ | 3.83 | 4.25 | 3.00 | 77.98 | 76.23 | 62.10 | 70.97 | 69.15 | 74.45 | 72.84 | 77.15 | 60.14 | 63.69 | 62.00 |
| | MAE | C-CVA | ✗ | 5.14 | 4.38 | 6.67 | 78.58 | 76.81 | 62.27 | 70.41 | 69.55 | 74.71 | 72.72 | 76.89 | 56.43 | 59.64 | 59.06 |
| ResEnc-L | MAE | ✗ | ✓ | 2.53 | 3.13 | **1.33** | 80.05 | 78.18 | 62.31 | 70.37 | 69.82 | 74.84 | 72.80 | 76.70 | 61.09 | 64.93 | 62.67 |
| | MAE | CVA | ✓ | **2.47** | 2.88 | 1.67 | 78.97 | 77.09 | 62.35 | 70.94 | 69.75 | 74.85 | 72.84 | 77.02 | 62.02 | 64.46 | 62.62 |
| | MAE | C-CVA | ✓ | 2.72 | **1.75** | 4.67 | 79.65 | 77.90 | 62.43 | 70.30 | 69.94 | 75.18 | 72.86 | 77.24 | 57.17 | 62.48 | 61.60 |
| | | | Range | | | | 2.70 | 2.61 | 2.02 | 2.28 | 1.67 | 1.83 | 0.64 | 0.74 | 5.60 | 5.28 | 3.78 |
| | AE | ✗ | ✗ | 5.03 | 6.38 | 2.33 | 76.05 | 73.35 | 51.92 | 58.43 | 63.35 | 69.93 | 71.44 | 75.90 | 56.09 | 61.79 | 60.51 |
| | MAE | ✗ | ✗ | 6.31 | 6.13 | 6.67 | 77.18 | 74.98 | 52.70 | 59.01 | 65.82 | 72.58 | 71.41 | 75.44 | 54.80 | 58.75 | 58.26 |
| | MAE | CVA | ✗ | 4.64 | 4.63 | 4.67 | 77.18 | 75.00 | 53.58 | 59.95 | 65.96 | 72.73 | 71.56 | 75.54 | 55.83 | 59.17 | 58.38 |
| | MAE | C-CVA | ✗ | 2.97 | 2.63 | 3.67 | 77.40 | 75.01 | 53.42 | 59.36 | 67.21 | 73.99 | 71.82 | 76.14 | 55.84 | 59.25 | 58.84 |
| Primus-M | MAE | ✗ | ✓ | **2.50** | 3.25 | **1.00** | 77.18 | 75.36 | 54.87 | 61.74 | 65.82 | 72.65 | 71.78 | 75.85 | 58.55 | 62.42 | 61.60 |
| | MAE | CVA | ✓ | **2.49** | **2.13** | 3.21 | 77.33 | 75.14 | 54.78 | 61.60 | 66.48 | 73.27 | 71.86 | 76.10 | 56.13 | 59.10 | 59.31 |
| | MAE | C-CVA | ✓ | 4.03 | 2.88 | 6.33 | 78.07 | 75.77 | 54.44 | 60.92 | 66.20 | 72.91 | 71.66 | 75.61 | 55.55 | 58.85 | 57.69 |
| | | | Range | | | | 2.01 | 2.42 | 2.95 | 3.31 | 3.86 | 4.07 | 0.45 | 0.69 | 3.75 | 3.67 | 3.90 |

**Results and Analysis** Table 1 compares Auto Encoder and MAE baselines, Contrastive MAE, and our alignment-based variants (CVA and C-CVA) across both architectures, with and without a global contrastive term. Alignment-based consistency consistently improves segmentation over MAE baselines. For *ResEnc-L*, CVA with contrastive regularization achieves the best rank (1.75), while for *Primus-M*, C-CVA alone performs best (2.63). These results indicate that local alignment sharpens spatial features, with the effect of the contrastive term depending on architecture. For classification on ABIDE II, Contrastive MAE remains strongest, showing that global contrastive objectives favour class-level separability, while CVA variants trade some global discriminability for local consistency.

**Discussion.** Local consistency improves segmentation, while contrastive objectives favour classification. Combining both CVA and the contrastive signal yields the best overall balanced pre-training strategy enabling robust and transferable representations across architectures and tasks.

## Potential Negative Societal Impact

While our framework aims to improve self-supervised learning for 3D medical imaging, it inherits risks associated with large-scale pre-training. The *OpenMind* dataset, though diverse, may still underrepresent certain populations or imaging protocols, potentially leading to biased feature representations that could diminish model performance for at-risk or underrepresented groups. Moreover, large-scale pre-training carries sustainability concerns due to substantial computational and energy costs. Although our two-stage protocol reduces total compute from 112 to 30 GPU-days across all ablations, this remains nontrivial. Future work should explore more data-efficient and equitable pre-training strategies that minimize environmental impact while ensuring fair generalization across demographic and clinical subgroups.

## Acknowledgments and Disclosure of Funding

This publication is part of the project ROBUST: Trustworthy AI-based Systems for Sustainable Growth with project number KICH3.LTP.20.006, which is (partly) financed by the Dutch Research Council (NWO), Siemens Healthineers, and the Dutch Ministry of Economic Affairs and Climate Policy (EZK) under the program LTP KIC 2020-2023.

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
