# OpenReview forum: "Consistent View Alignment Improves Foundation Models for 3D Medical Image Segmentation"
_EurIPS.cc/2025/Workshop/MedEurIPS — EurIPS 2025 Workshop MedEurIPS Submission_

### Official Review · Reviewer_iaYS · 2025-10-27
**The authors introduces a self-supervising learning framework to improve representation for 3D medical image segmentation. The idea is to apply a consistency loss only to features corresponding to the known overlapping region. The results showed that combining local and global objectives provides most balanced pretraining strategy.**

**Rating:** 7
**Confidence:** 3

**Review:**

**Strengths**
- Clear motivation, also demonstrated by the figure regarding the loosely correlated view problem.
- Sound methodology and results section.
- Evaluation of the downstream task (incl. segmentation and classification)

**Weakness**
- Though nothing particular in context to the workshop's guidelines, I see a missing ablation discussion (or reason for choice made) for the fixed overlap ratio (40-80%).
- The classification performance tradeoff: though mentioned in the discussion, but no investigation on "why".

---

### Official Review · Reviewer_cR27 · 2025-10-29
**Overall, the paper is well written, and has an original contribution; but the results are not entirely convincing, and clarity should be improved in the method section.**

**Rating:** 6
**Confidence:** 4

**Review:**

Overall, the paper is well written, and has an original contribution;
but the results are not entirely convincing, and
clarity should be improved in the method section.

The segmentation improvement from local feature alignement
is asserted using the rank metric, but looking at DSC metrics on
individual datasets, the gap appears negligible. Authors should
consider statistical testing. Since the baseline contrastive model
outperforms the consistency model on classification, it may not
be justified to pretrain a segmentation-specific foundational model.

Pros: Simplicity of the idea; well-posed problem in introduction and Figure 1;
         Most sections well written, with clear figures.
         Improvement of the segmentation results, although not statistically demonstrated.

Cons: Inconsistencies in method section (Lconsis, Lcon, Lrecon, undefined);
           No statistical analysis to support claims;
           1st sentence of the abstract not clear enough.

---

### Decision · Program_Chairs · 2025-10-31

**Decision:**

Accept (Poster)

**Comment:**

Both reviewers find the paper well motivated and methodologically sound, introducing an original idea for improving self-supervised representation learning through consistent view alignment.